# BASHY Dye Platform Enables the Fluorescence Bioimaging of Myelin Debris Phagocytosis by Microglia during Demyelination

**DOI:** 10.3390/cells10113163

**Published:** 2021-11-13

**Authors:** Maria V. Pinto, Fábio M. F. Santos, Catarina Barros, Ana Rita Ribeiro, Uwe Pischel, Pedro M. P. Gois, Adelaide Fernandes

**Affiliations:** 1Research Institute for Medicines (iMed.ULisboa), Faculdade de Farmácia, Universidade de Lisboa, 1649-003 Lisbon, Portugal; mariagvp2009@gmail.com (M.V.P.); fmfsantos1987@gmail.com (F.M.F.S.); catarinabarros@campus.ul.pt (C.B.); ar.ribeiro@campus.fct.unl.pt (A.R.R.); 2CIQSO (Centro de Investigación en Química Sostenible)—Centre for Research in Sustainable Chemistry and Department of Chemistry, University of Huelva, 21071 Huelva, Spain; uwe.pischel@diq.uhu.es; 3Department of Pharmaceutical Sciences and Medicines, Faculdade de Farmácia, Universidade de Lisboa, 1649-003 Lisbon, Portugal

**Keywords:** multiple sclerosis, demyelination, microglia phagocytosis, myelin debris, BASHY, in vivo imaging

## Abstract

Multiple sclerosis (MS) is a demyelinating disease of the central nervous system that is characterized by the presence of demyelinated regions with accumulated myelin lipid debris. Importantly, to allow effective remyelination, such debris must be cleared by microglia. Therefore, the study of microglial activity with sensitive tools is of great interest to better monitor the MS clinical course. Using a boronic acid-based (BASHY) fluorophore, specific for nonpolar lipid aggregates, we aimed to address BASHY’s ability to label nonpolar myelin debris and image myelin clearance in the context of demyelination. Demyelinated ex vivo organotypic cultures (OCSCs) and primary microglia cells were immunostained to evaluate BASHY’s co-localization with myelin debris and also to evaluate BASHY’s specificity for phagocytosing cells. Additionally, mice induced with experimental autoimmune encephalomyelitis (EAE) were injected with BASHY and posteriorly analyzed to evaluate BASHY^+^ microglia within demyelinated lesions. Indeed, in our in vitro and ex vivo studies, we showed a significant increase in BASHY labeling in demyelinated OCSCs, mostly co-localized with Iba1-expressing amoeboid/phagocytic microglia. Most importantly, BASHY’s presence was also found within demyelinated areas of EAE mice, essentially co-localizing with lesion-associated Iba1^+^ cells, evidencing BASHY’s potential for the in vivo bioimaging of myelin clearance and myelin-carrying microglia in regions of active demyelination.

## 1. Introduction

Multiple sclerosis (MS) is the primary chronic demyelinating disease of the central nervous system (CNS) and the leading cause of non-traumatic disability in young adults [1]. Current evidence suggests that MS conditions are closely tied to an immune system dysregulation that leads to myelin sheath degradation into nonpolar lipid fragments within MS-characteristic demyelinated plaques [2]. Importantly, myelin debris contains toxic lipids and myelin-associated proteins known to inhibit both neurite growth and the differentiation of oligodendrocyte precursor cells into mature/myelinating ones. Thus, efficient clearance of such debris by microglia is one of the important processes that need to occur to promote efficient remyelination [3,4,5,6,7]. Otherwise, continuous demyelination alongside incompetent/inexistent myelin removal causes non-treatable neurodegeneration, which clinically translates into progressive disability at motor, sensitive, and cognitive levels, having severe impacts on the patients’ quality of life [8,9]. Given the complex etiology of MS, its currently practiced treatments rely on disease-modifying therapies to reduce immune pathogenesis, ease symptomatology, and slow down MS progression. Therefore, assessing microglial ability to phagocytose myelin with highly sensitive tools is of focal interest to better comprehend, diagnose, and monitor clinical MS conditions and to discover therapeutic approaches that promote disease recovery through remyelination.

Non-invasive brain imaging tools have emerged as fundamental aspects in the assessment of MS diagnosis, pathological monitoring, and treatment response. Indeed, magnetic resonance imaging (MRI), although lacking specificity for MS pathology, is currently in use for the identification of myelin alterations [10,11]. Moreover, positron emission tomography (PET), using myelin-specific tracers, is a useful tool that locates demyelinated lesions over MS disease course, fairly correlating with histological analysis [11,12,13]. Moreover, the available in vivo imaging systems further evolved to assess inflammatory active demyelinating lesions through gadolinium-enhanced MRI analysis [14] and by using translocator protein-18 (TSPO)-specific PET ligands [15,16]. However, despite these technological advances, the process by which microglia perform the clearance of myelin debris is poorly understood. This is foremost due to the lack of imaging techniques capable of accurately identifying microglia and microglial phenotypes [17,18] among demyelinating lesions in live cells, as all the currently available methods are applicable only to ex vivo samples.

In line with this, we recently developed a modular fluorescent platform based on boronic acid salicylidenehydrazone complexes (BASHY) and observed that these hydrophobic dyes could distinguish nonpolar lipid structures from other lipid frameworks, such as plasmatic membranes [19]. Therefore, given myelin’s great enrichment in lipids [20], we further conceived that BASHY dyes could be engineered to label these hydrophobic myelin fragments and not intact myelin sheaths. Here, we exploit the use of a BASHY dye as an improved fluorophore with increased affinity for myelin debris, which enabled us to target myelin-phagocytosing cells with great efficacy in ex vivo demyelinating samples. Additionally, we proved BASHY’s excellent stability once inside microglial cells and its great selectivity for activated lipid-rich phagocytosing cells, the ones that are expected to be found in active MS demyelinating lesions. Finally, as a preliminary approach to study the efficiency of BASHY in vivo, we used the in vivo model of MS, the experimental autoimmune encephalomyelitis (EAE), most commonly employed in pre-clinical studies. Not only does the EAE model closely resemble most of the key features of MS (e.g., demyelination, inflammation, glial reactivity, and axonal loss), but the presence of accumulated myelin debris [21] and foamy phagocytes within demyelinated plaques in EAE-induced animals has already been reported [22]. Indeed, by using BASHY in EAE-challenged mice, we demonstrated BASHY’s potential as a novel fluorescent probe to study myelin clearance by microglia in the context of demyelination and ultimately during the MS pathogenesis.

## 2. Materials and Methods

### 2.1. BASHY Synthesis and Characterization

Method A: In a round-bottomed flask, Schiff base ligand 1” (0.1 millimole (mmol)) and phenylboronic acid (0.1 mmol) were mixed in acetonitrile (1 milliliter (mL)) at 80 °C for 2 h. Then, volatiles were evaporated under reduced pressure, and BASHY test dye (td)**1** was obtained as an orange solid in near quantitative yield (99%).

Method B: In a round-bottomed flask, equimolar amounts (0.1 mmol) of salicylhydrazone 5, phenylboronic acid, and the corresponding phenylglyoxylic acid derivative were mixed in acetonitrile (1 mL) at 80 °C for 2 h. Then, volatiles were evaporated under reduced pressure, and the crude mixture was purified via thin layer chromatography using dichloromethane as eluent. BASHY td**2–4** were obtained as orange to red solids in good yields (70–82%).

Additional synthesis description and structural characterization data can be found in the Appendix A.

### 2.2. Ex Vivo Demyelinating Model

To evaluate BASHY staining following a demyelinating event, we used an ex vivo model of demyelination as previously described [23]. In short, cerebella from postnatal day 10 (P10) Wistar rats were isolated in phosphate-buffered saline (PBS), and sagittal slices of 400 micrometers (μm) were obtained using a McIlwain tissue chopper. Four slices from different animals were placed into each membrane culture insert, with 0.4 μm pores (BD Falcon, #353493, Lincoln Park, NJ, USA), and the inserts were placed in 6-well cell culture plates kept at 37 °C, in 5% CO_2_ conditioned atmosphere for 7 days in vitro (DIV), to allow the recovery and significant myelination of the organotypic slice cultures (OCSCs) [24]. For 3 DIV, we maintained OCSCs with culture medium (1 mL per well) consisting of 50% minimal essential media (MEM) (Gibco, Life Technologies, Inc., Grand Islands, NE, USA), 25% of both heat-inactivated horse serum (Gibco) and Earle’s balanced salt solution (EBSS, Gibco), 6.5 milligrams (mg)/mL glucose, 36 millimolar (mM) HEPES (Biochrom AG, Berlin, Germany), and 1% of both L-glutamine (Sigma-Aldrich, St. Louis, MO, USA) and antibiotic/antimycotic (Sigma-Aldrich). At 4 DIV, to improve neuronal viability, the culture media were totally replaced by serum-free media (1 mL per well), containing 98% Neurobasal-A (NB) (Gibco) supplemented with 2% B-27 (Gibco), 1% L-glutamine, 36 mM glucose, 1% of antibiotic/antimycotic, and 25 mM HEPES. Half of the culture media was renewed every day. Following 7 DIV, OCSCs were exposed to lysophosphatidylcholine (LPC) (0.5 mg/mL, in NB) for 18 h to induce demyelination, after which, the media were completely replaced by NB fresh medium for a recovering period of 30 h. At 18 h and 48 h post-LPC induction, induced and control OCSCs were either stored in RiboZolTM reagent at −80 °C for further RNA extraction or fixed in paraformaldehyde (PFA, 4% (*w*/*v*) in PBS) for 1 h for future immunohistochemistry assays and staining analysis with BASHY, Lysotracker, and Nile Red.

### 2.3. Primary Culture of Microglia

Rat microglia were isolated from mixed glial cultures prepared from P10 Wistar rats as previously described by us [25]. Briefly, brains were collected (in DMEM-Ham’s F-12 solution), and meninges were removed. Afterward, we homogenized the cortex by mechanical fragmentation and passed the cell suspension sequentially through steel screens of 230 and 104 μm pore size. Cells were collected by centrifugation (1200 rpm for 10 min) and resuspended in glia-conditioned medium: DMEM-Ham’s F-12 medium supplemented with 10% fetal bovine serum (FBS), 1 mM sodium pyruvate, 2 mM L-glutamine, 1% nonessential amino acids, and 1% antibiotic/antimycotic solution. Finally, cells (4 × 10^5^ cells/cm^2^) were plated on uncoated 6-well tissue culture plates (Corning Costar Corp., Cambridge, MA) and maintained at 37 °C in a humidified atmosphere of 5% CO^2^. Microglia were isolated as previously described [26]. After 21 days in mixed culture in vitro—to achieve maximal yield—microglia were obtained by mild trypsinization with a trypsin–EDTA solution (1:3 in DMEM-F12) for 45 min at 37 °C, which resulted in the detachment of an upper layer of cells containing all the astrocytes. Microglial cells remained attached to the bottom of the well. The medium containing the layer of detached cells was removed and replaced with the initial mixed glial-conditioned medium. Twenty-four hours later, the isolated microglia were detached from the bottom of the well after trypsinization with trypsin–EDTA solution and cultured in 96-well culture plates (1000 cells/well) for another 24 h, after which cells were used for the phagocytosis assay.

### 2.4. Culture of Human CHME3 Microglia Cell Line 

Human CHME3 microglial cells were cultured in T75 culture flasks in DMEM supplemented with 10% FBS, 2% antibiotic/antimycotic (Sigma-Aldrich), and 1% L-glutamine (Sigma-Aldrich) in a humidified atmosphere containing 5% CO_2_ at 37 °C. Medium was changed every 2–3 days. Prior to the incubation with myelin debris, cells were seeded onto 6-well non-coated plaques for 24 h at a final concentration of 1 × 10^5^/mL.

### 2.5. Culture of Immortalized Human Fetal 10B1 Astrocytic Cell Line

Human 10B1 astrocytic cells were cultured in T75 culture flasks in DMEM supplemented with 10% FBS, 1% penicillin/streptomycin, 1% L-glutamine, and 0.5% gentamicin in a humidified atmosphere containing 5% CO_2_ at 37 °C. Medium was changed every 2–3 days. Prior to the incubation with myelin debris, cells were seeded onto 6-well non-coated plaques for 24 h at a final concentration of 1 × 10^5^/mL.

### 2.6. Experimental Autoimmune Encephalomyelitis and BASHY Injection

To perform EAE studies, female C57BL/6 mice aged 8–10 weeks were acquired from Instituto Gulbenkian Ciência. All animal procedures were performed in accordance with the guidelines of the Portuguese national authority for animal experimentation, Direção Geral de Alimentação e Veterinária, to minimize their suffering. For the chronic EAE model, mice were induced using a commercial kit (Hooke laboratories, Lawrence, MA, USA) according to the manufacturer’s instructions. On day 0, animals were immunized by a subcutaneous injection in the upper and lower back (100 microliters (μL) per site) with myelin oligodendrocyte glycoprotein 35–55 (MOG35–55) peptide emulsified in complete Freud’s adjuvant (CFA). To achieve full immunization, mice were administered intraperitoneally with pertussis toxin (PTx) in PBS (100 μL per animal), both on the first day of immunization and 24 h after.

C57BL/6 mice were divided into four different groups: (1) control group receiving BASHY retro-orbital intravenous injection (I.V.); (2) control group receiving BASHY intraperitoneal injection (I.P.); (3) EAE-challenged group receiving BASHY I.V.; and (4) EAE-challenged group receiving BASHY I.P. BASHY was injected (100 μL (1 mM)/20 g body weight) 24 h before and again 1 h before animals were sacrificed. Afterward, mice were anaesthetized with a non-lethal dose of isoflurane and intracardially perfused through the left heart ventricle with PBS using a peristaltic pump. Mice cortex, cerebellum, and spinal cord were (1) fixed in PFA at 4 °C, then cryoprotected with 40% sucrose in PBS, and, further, snap-frozen in TissueTek O.C.T. compound (Sakura Finetek Europe, Alphen aan den Rijn, the Netherlands) for immunohistochemistry or (2) collected and dissociated to perform flow cytometry analyses.

### 2.7. Fluorescent Probe Staining Assay 

Following fixation, membranes containing OCSCs were cut out from the insert. Control and LPC-induced OCSCs were incubated with BASHY molecules 1–4 solubilized in acetonitrile (100 μL, 5 micromolar (μM) in PBS) for 20 min at room temperature (RT). In parallel experiments, BASHY molecule 2 (5 μM) was co-incubated with LPC after 7 DIV. OCSCs were cut out from the insert and (1) incubated with Nile Red (1:1000, Sigma) or Lysotracker (1:20,000, TermoFisher) for 30 min at 37 °C prior to fixation.

In all staining procedures, OCSCs were mounted using Fluoromount-G (Southern Biotech, Birmingham, AL, USA) for fluorescence/confocal microscopy. Fluorescent images were acquired using a Leica DMi8-CS inverted microscope with Leica LAS X software. To measure the % of BASHY-stained area, a threshold was defined for the z-stacks that corresponds to a minimum intensity due to specific staining above background signals. With the established threshold value, the % of BASHY labeling was automatically calculated per analyzed area using Fiji (Fiji Is Just ImageJ). In parallel, the number of BASHY-stained (green) particles (>10 μm^2^, to avoid background signal) was automatically counted using Fiji (Fiji Is Just ImageJ). Finally, BASHY co-localization with Lysotracker and Nile Red was analyzed by measuring and overlapping the two signaling intensity profiles. The signal intensity profiles were calculated using Fiji (Fiji Is Just ImageJ) in at least 3 individual z-stacks per field of view after establishing a threshold value. 

### 2.8. Myelin Debris Isolation and Staining 

Myelin debris from P10 rats was isolated using a sucrose density gradient method (0.32 Molar (M) and 0.85 M) at 100,000× *g*, as described [27]. Myelin debris was collected from the interface of the two sucrose densities and further resuspended in 200 μL of BASHY 2 per 100 μL myelin debris pelleted [27]. Myelin-labeled debris (100 mg/mL) was used to stimulate microglial cells for the in vitro phagocytic assay.

### 2.9. Myelin Debris Phagocytosis Assay

Microglial phagocytosis of BASHY-labeled myelin debris was assayed according to a previously published protocol with minor modifications [27]. We added 1 μL of myelin-labeled debris (100 mg/mL) to each well containing the previously isolated microglial cells to a final concentration of 1 mg/mL. Then, non-phagocytosed myelin debris was washed out three times with PBS. Microglial primary cells were fixed with 4% (*w*/*v*) PFA in PBS for 30 min at RT to perform future immunocytochemistry. In parallel, 25 μL of (1) non-stained (negative control) and (2) BASHY-stained myelin debris was added to each well containing human CHME3 microglial cells and 10B1 astrocytes to a final concentration of 1 mg/mL. Cells with myelin debris were incubated for 1 h at 37 °C. Afterward, non-phagocytosed myelin debris was washed out three times with PBS, and cells were collected to perform flow cytometry analysis. 

### 2.10. Immunostaining Procedures

For immunostaining of OCSCs, after fixation, membranes containing the OCSCs were cut out form the insert; placed into glass slides; and blocked for three hours at RT with blocking solution containing 2% heat-inactivated horse serum (Gibco), 10% fetal bovine serum (Biochrom), 1% bovine serum albumin (BSA, Sigma-Aldrich), 0.25% Triton X-100 (Roche Diagnostics, Indianapolis, IN, USA), and 1nM HEPES in Hank’s balanced salt solution (HBSS, Gibco). Afterward, we incubated OCSCs with primary antibody (diluted in blocking solution) for approximately 48 h at 4 °C. We used the following antibodies: myelin basic protein (Mbp, 1:200, BioRad) for mature oligodendrocytes/compact myelin sheaths, Qd9 (1:100, Abnova) for degraded myelin, glial fibrillary acidic protein (Gfap, 1:100, Novocastra) for astrocytes, ionized calcium-binding adapter molecule 1 (Iba1, 1:250, WAKO), arginase1 (Arg1, 1:50, Santa Cruz) for anti-inflammatory microglia, and inducible nitric oxide synthase (iNOS, 1:100, BD Biosciences) for pro-inflammatory microglia. Then, OCSCs were washed three times with 0.01% Triton X-100 in PBS for 20 min each, under shaking at RT before being probed overnight at 4 °C with the following secondary fluorescent antibodies: anti-rabbit Alexa Fluor 594 and 405, anti-rat Alexa Fluor 594, anti-mouse Alexa Fluor 488 and 647, and anti-goat Alexa Fluor 594 (1:500, in blocking solution). OCSCs were washed three times in the same conditions and incubated with 4′,6-diamidino-2-phenylindole (DAPI) (1:1000 in PBS) for 5 min to stain total cell nuclei. After 1 wash with PBS, OCSCs were finally mounted using Fluoromount-G for fluorescence/confocal microscopy. To measure the areas of co-localization of BASHY with Mbp, Qd9, Gfap, and Iba1, a threshold was defined for the z-stacks that corresponds to a minimum intensity due to specific staining above background signals. With the established threshold value, the area of BASHY labeling was automatically calculated in 3 regions of interest (ROI, constituting an area of 4.0 × 10^5^ μm^2^) using Fiji (Fiji Is Just ImageJ). Quantifications of Iba1^+^ cells were performed in 3 ROI within 3 sections/cerebellar white matter regions. The data are presented as the number of cells counted per ROI, with each ROI constituting an area of 3.45 × 10^5^ μm^2^.

To observe BASHY-labeled debris in primary microglial cultures, standard immunocytochemistry was performed. Fixed microglial cells incubated with BASHY-labeled myelin debris were permeabilized with 0.2% Triton X-100 (Roche Diagnostics, Indianapolis, USA) in PBS for 20 min at RT and blocked with blocking solution containing 3% (*w*/*v*) BSA (Sigma-Aldrich) in PBS for 30 min at RT. Then, they were incubated overnight at 4 °C with the primary antibody anti-Iba1 diluted in 1% (*w*/*v*) BSA in PBS solution. Following 3 washes with PBS, cells were incubated with secondary antibody anti-rabbit Alexa Fluor 594 diluted in the same solution for 2 h at RT. Again, cells were washed three times in the same conditions, stained with DAPI for 5 min, washed 1 time with PBS, and finally mounted using Fluoromount-G for fluorescence/confocal microscopy.

For immunostaining of slides from control and EAE-challenged animals, frozen brain sections with 20 μm thickness were collected on Superfrost Plus glass slides, defrosted at RT, and post-fixed in 4% PFA for 10 min. After 3 washes (10 min each with PBS), sections were permeabilized with 0.25% Triton X-100 in PBS for 10 min and then incubated with blocking solution containing 5% bovine serum albumin, 5% fetal bovine serum, and 0.1% Triton X-100 in PBS solution for 1 h at RT. Next, we incubated sections with primary antibody (diluted in blocking solution) for approximately 48 h at 4 °C. We used the following antibodies: anti-Mbp and anti-Iba1. Afterward, sections were washed three times for 10 min each with PBS and incubated with the appropriate secondary fluorescent antibodies (anti-rat Alexa Fluor 594 and anti-rabbit Alexa Fluor 594, diluted in blocking solution) for approximately 2 h at RT. Finally, sections were washed three times for 10 min each with PBS and incubated with DAPI for 5 min as indicated above. sections were washed again three times for 5 min each with PBS and then mounted as above for fluorescence/confocal microscopy.

### 2.11. Semi-Quantitative Real-Time PCR

Total cytoplasmic RNA was extracted from OCSCs at both time points (18 h and 48 h post-LPC), using RiboZolTM reagent method following the manufacture’s guidelines (VWR Life Science, USA). RNA concentration and purity were quantified using Nanodrop ND-100 Spectrophotometer (NanoDrop Technologies, Wilmington, DE, USA). After extraction and quantification, RNA samples were reversely transcribed into complementary DNA (cDNA) using Xpert cDNA synthesis Mastermix kit (GRiSP) according to the manufacturer’s guidelines. Quantitative real-time PCR (qRT-PCR) for cDNA amplification was performed on a 7300 Real-Time PCR detection system (Applied Biosystem, Madrid, Spain) using an Xpert Fast SYBR Mastermix (GRiSP) kit under optimized conditions of 50 °C for 2 min and 95 °C for 10 min, followed by 40 cycles at 95 °C for 5 s and 62 °C for 30 s. The PCR was performed in 384-well plates with each sample being performed in duplicate. We used the β-actin gene as an endogenous control to normalize the expression level of first-line cytokines: interleukin (IL)-1β, F-5′ CAGGCTCCGAGATGAACAAC 3′ and R-5′ GGTGGAGAGCTTTCAGCTCATA 3′; tumor necrosis factor (TNF)-α, F-5′ TACTGAACTTCGGGGTGATTGGTCC 3′ and R-5′ CAGCCTTGTCCCTTGAAGAGAACC 3′; and Interleukin (IL)-10, F-5′ ATGCTGCCTGCTCTTACTGA 3′ and R-5′ GCAGCTCTAGGAGCATGTGG 3′. For semi-quantitative analysis of the transcription levels of our genes of interest, we used the 2^−ΔΔCT^ comparative method. 

### 2.12. Flow Cytometry 

To each well of human CHME3 microglial cells and 10B1 astrocytes, 450 μL of trypsin was added for 5 min at 37 °C to detach the cells. We then added another 50 μL of FBS to each well and, afterward, collected the detached cells into an Eppendorf for centrifugation at 500× *g* for 5 min. Next, cells were fixed with 1% PFA in PBS for 10 min at RT and then resuspended and centrifuged again at 500× *g* for 5 min. The pellet was finally resuspended in 300 μL of FACS buffer (2% FBS in PBS). A total of 50,000 cells were analyzed on a Cytek^®^ Aurora flow cytometer.

Samples of cortex, cerebellum, and spinal cord from all animals were dissociated by mechanical fragmentation for flow cytometry analysis as described in [28] with minor modifications. Briefly, the dissociated tissue was incubated with 1 mL of collagenase (1 mg/mL, in HBSS without Ca2^+^ and Mg2^+^) for 30 min at 37 °C with regular agitation. Afterward, 14 mL of FACS buffer was added, and the dissociated tissue was centrifuged (1200 rpm for 10 min), after which, the supernatant was discarded, and the pellet was resuspended in 1 mL of FACS buffer. Next, we passed the cell suspension through a steel screen of 73 μm pore size, and cells were collected by centrifugation (1200 rpm for 10 min). Finally, the pellet was resuspended in 1 mL of FACS buffer. For cell staining, we collected 100 μL of cells, and they were centrifuged at 500× *g* for 5 min. The pellet was incubated with 100 μL of blocking solution (5% FBS in Tris-buffered saline) for 10 min before another centrifugation (500× *g* for 5 min). Again, the supernatant was removed, and cells were stained with 100 μL of antibody anti-CD80 (PE/Cy5) (1:100 in blocking solution) for 30 min at RT. Lastly, cells were centrifuged at 500× *g* for 5 min, and the pellet was resuspended in 600 μL of FACS buffer. A total of 30,000 cells were analyzed on a Guava easyCyte 5HT flow cytometer (Guava Nexin^®^ Software module, Millipore, Burlington, MA. USA). 

### 2.13. Statistical Analysis 

All results are presented as mean ± SEM. Differences between two groups were determined by the two-tailed t-test performed on the basis of equal and unequal variance or by one-way ANOVA with Tukey post-test for multiple comparisons, using GraphPad PRISM 5.0 (GraphPad Software, San Diego, CA, USA), as appropriate. The *p*-values of *p* < 0.05, *p* < 0.01, and *p* < 0.001 were considered as being statistically significant.

### 2.14. Safety Statement

No unexpected or unusually high safety hazards were encountered.

## 3. Results

### 3.1. Development of BASHY Probe to Detect Fragmented Myelin

BASHY dyes (see structures of the herein employed compounds in Figure 1A) constitute an innovative and versatile platform of fluorophores that can be tailored for the specific needs of bioimaging applications [19,29,30]. These dyes consist of a salicylidenehydrazone ligand that is conformationally locked by reaction with a boronic acid component leading to a rigid non-planar structure. The photophysical properties (see Appendix A for the td**1**–**4** are strongly dependent on the push–pull character that is propagated along the conjugated salicylidenehydrazone axis, enabling flexible fine tuning through the choice of donor/acceptor substitution. Remarkably, the dyes show a pronounced light-up effect when transferred from a polar to an apolar environment [19]. This functionally very attractive aspect is ascribed to the push–pull character of BASHY, which is paired with cyanine-like behavior [30]. The relatively low stability of td**1** against hydrolysis (half-life less than 10 min) can be alleviated by proper substitution at the electrophilic imine carbon, as shown for the dye td**2** (half-life of ca. 3.5 h). On extension of the π-conjugation of td**2**, leading to the cyanine-like chromophores td**3** and td**4**, the hydrolytic stability can be further improved for second-generation BASHY dyes (only 5% hydrolysis after 3 h in 10 mM phosphate-buffered saline at pH 7.4 as reaction medium) [30]. The stability data are summarized in Appendix A.

With the intention to pre-test the four probe candidates regarding the specific challenge of myelin debris bioimaging, we used td**1***–***4** in an ex vivo model of OCSCs, in which demyelination was induced with LPC (Figure 1B). Demyelination starts with myelin decompaction, followed by membrane vesiculation and degradation into lipid-rich myelin debris within the extracellular space [31]. Therefore, BASHY labeling was expected to provide bioimaging evidence for this accumulation of myelin debris. In a first approach, 48 h post-LPC induction, the OCSCs were fixed and stained with BASHY td**1–4** (Appendix A).

Among the tested dyes, the green-emitting td**2** presented superior performance with a low background emission and high staining level following LPC-induced demyelination (Appendix A). Encouraging results were also obtained for the π-extended dyes td**3** and td**4** (Appendix A). However, as expected from their increased conjugated character, the fluorescence signals of td**3** and td**4** were red-shifted in comparison to td**2**. This constitutes a potential problem for the concomitant detection of co-localized markers. Notably, the more red-shifted emission also comes at the expense of smaller fluorescence quantum yields and brightness, as reasoned with a faster non-radiative deactivation. In addition, the “fluorescence contrast” between polar and apolar environments (i.e., the light-up effect) is less pronounced than for td**2**; see Appendix A. Combining the photophysical evidence and the ex vivo bioimaging results prompted us to conclude that td**2** (from here on simply referred to as BASHY) is the ideal lead compound for in-depth biological studies on the demyelination process (see below).

As shown in Figure 1C, we proved that the fluorescent area of BASHY was increased in LPC-induced OCSCs (2.88-fold, *p* < 0.05), which feature an enhanced accumulation of myelin debris, when compared to the controls. The same was observed when counting the number of green particles in control and LPC-induced OCSCs (2.88-fold, *p* < 0.05) (Figure 1D), which is suggestive of BASHY labeling of demyelinated structures. 

### 3.2. Detection of Myelin-Enriched Macrophage/Microglia Using BASHY Molecules

Next, we decided to assess the cellular localization of BASHY in order to ascertain its ability to identify degraded myelin structures or its selectivity for the myelin-phagocytosing cells, macrophages, and/or microglia, the ones responsible for the clearance of accumulated myelin lipid fragments. Therefore, fixed OCSCs were immune stained for mature oligodendrocytes/intact myelin (Mbp), degraded/non-compact myelin (Qd9), astrocytes (Gfap), and macrophages/microglia (Iba1) and posteriorly stained with BASHY. 

We observed that upon demyelination, BASHY fluorescence accompanies the formation of some early vesiculated structures concomitant with the loss of myelin compaction. Indeed, there was a slight increase in the area of co-localization with Qd9 in OCSCs incubated with LPC (Figure 2A) when compared with control OCSCs, which markedly showed a reduced BASHY staining (Figure 2A and Appendix A). Moreover, although we observed some co-localization with astrocytes, it was not as evident as with microglia/macrophages (Figure 2A and Appendix A), as astrocytes have a decreased capacity to phagocytose myelin debris (Appendix A) when compared to microglia (Appendix A). As a result of cell activation following mechanical injury during slice preparation, we observed the presence of BASHY staining that was co-localizing with Iba1 in control OCSCs. However, this was again not as evident as in demyelinated OCSCs. Our results clearly show that BASHY fluorescence is strongly confined to Iba1-positive macrophage/microglial cells following demyelination with LPC (Figure 2A and Appendix A), which sustains our hypothesis that BASHY can label myelin debris and, by doing so, accompanies the destruction of myelin and its accumulation inside myelin-phagocytosing cells. 

To assure BASHY affinity for myelin debris, as well as its internalization by myelin-phagocytosing cells, we collected myelin debris from rat brains (day 10 postnatal, P10), stained it with BASHY, and posteriorly incubated a primary culture of microglia with such BASHY-labeled myelin debris (MD-BASHY) (Figure 2B). Consistent with the ex vivo results, we found MD-BASHY (green) clearly accumulated within microglia (Iba1, red), strongly indicating that BASHY maintains its selectivity after in vivo internalization by microglial cells (Figure 2C). 

Furthermore, at the subcellular level, it is described that after cell internalization, myelin lipids accumulate within lysosomal vesicles before being transported to the endoplasmic reticulum and posteriorly stored into lipid droplets or effluxed from the cell (Figure 2D) [32]. So, we next addressed if the BASHY dye could identify myelin debris along its intracellular metabolization path. For that, 48 h post-LPC, OCSCs were stained to observe BASHY (green), lysosomal vesicles (Lysotracker, red), and lipid droplets (Nile Red, Red). Interestingly, BASHY co-localizes both with Lysotracker (Figure 2D—upper panel and Appendix A) and Nile Red (Figure 2D—lower panel and Appendix A), which was further confirmed by the observed correlation of the BASHY fluorescence intensity profile with the one of each marker. 

Overall, we can conclude that the BASHY fluorophore accumulates in fragmented myelin structures, while showing low affinity for intact myelin layers, and maintains its binding upon microglia phagocytosis, accompanying lipid delivery into lysosomal structures as well as their accumulation in lipid droplets.

### 3.3. MD-BASHY Is Mainly Internalized by Amoeboid Microglia

Considering the previous results, we dissected BASHY internalization by myelin-phagocytosing macrophage/microglial cells, detailing the morphological changes upon demyelination: from a ramified morphology to a bushy and amoeboid activated cell (Figure 3A), these last ones resembling the characteristic “foamy or lipid-rich phagocytes” found in MS cases [33]**.** In our demyelinating ex vivo model, we observed a complete shift in macrophage/microglia morphology after LPC induction, with a reduction in ramified and bushy morphology but an accumulation of amoeboid cells in demyelinated OCSCs (Figure 3B and Appendix A). In accordance with the previous results, BASHY fluorescence increased over demyelination and followed macrophage/microglial morphological changes, thus essentially localizing with amoeboid phagocytic cells (*p* < 0.001 vs. bushy and ramified) (Figure 3B). 

Based on the observed preference of BASHY for amoeboid macrophages/microglia, we further assessed the differential phenotype of BASHY-bearing microglia as a challenging approach to discriminate pro- and anti-inflammatory cells. It is already known that following an inflammatory stimulus, such as demyelination, the activation process of microglia is associated with both morphological and functional changes. Besides evolving from a hyper-ramified morphology into a rod and amoeboid state (Figure 3A,B), cells undergo, at the same time, alterations in gene expression, transitioning from a homeostatic and surveillant status to various stages of activation and reactivity with opposing roles in disease progression [34]. Ex vivo studies on demyelination characterized initial activated microglia with a predominant pro-inflammatory and disease-potentiating state that further evolves into a more disease-resolving and anti-inflammatory phenotype [35]. So, we decided to evaluate the microglial phenotype at both 18 h and 48 h post-LPC incubation and to assess the specificity of BASHY for microglial differential profiles. This was carried out at each time point by immunostaining for Iba1; Arg1, a common marker for anti-inflammatory microglia; and for iNOS, significantly expressed by pro-inflammatory microglial cells [36]. As previously reported [35]**,** in our demyelinating ex vivo model, exposure to LPC induced differences in the cell phenotype when compared to the controls (Figure 3C). Although the majority of the cells were Arg1^+^ in all conditions, iNOS^+^ microglia were mainly observed at 18 h in demyelinated OCSCs (*p* < 0.001 vs. 18 h control OCSCs) (Figure 3C,D and Appendix A), indicating a shift in phenotype from an early pro-inflammatory to a less-inflammatory iNOS^-^ population over time (*p* < 0.01 vs. 18 h demyelinated OCSCs). Regarding BASHY selectivity, we observed the internalization of BASHY in both Arg1^+^/iNOS^+^ and Arg1^+^/iNOS^-^ amoeboid cells at each time point. Indeed, a quantitative analysis (Appendix A) showed that ~45% of the BASHY^+^ amoeboid cells were in a pro-inflammatory state (*p* < 0.01) 18 h after LPC induction, whereas at 48 h post-LPC exposure, ~95% of the BASHY^+^ amoeboid population consisted of an anti-inflammatory/Arg1-expressing phenotype (*p* < 0.001). 

We next evaluated the mRNA expression of IL-1β and TNF-α as pro-inflammatory cytokines and IL-10 as an anti-inflammatory one at each time point. Our results show that at 18 h post-LPC, both pro-inflammatory markers were upregulated (6.1-fold for IL-1β, *p* < 0.01 and 1.8-fold for TNF-α, *p* < 0.05), which was accompanied by the counteracting upregulation of the anti-inflammatory cytokine IL-10 (10.8-fold, *p* < 0.01) (Figure 3E). The inflammatory response was clearly reduced at 48 h post-LPC (1.2-fold for IL-1β and 0.7-fold for TNF-α, *p* < 0.01), which is consistent with both the previous reports [35] and our own immunohistochemistry results. The combined experimental evidence clearly supports that an early enhanced population of pro-inflammatory microglia/macrophages is progressively replaced by a more anti-inflammatory population (Figure 3C,D and Appendix A) and that BASHY accompanies the corresponding macrophage/microglial phenotypic changes over demyelination.

### 3.4. Detection of Myelin-Phagocytosing Microglia in In Vivo Demyelinating Lesions Using BASHY

In MS, myelin loss gives rise to the well-described lesioned areas or demyelinated plaques. These multiple lesions, highly prevalent in regions of the cerebellum (CB) [37], periventricular layers [38], and spinal cord (SC) [39] are rapidly filled microglial cells and peripheral macrophages to perform the removal of myelin-fragmented debris [2]. Given our results, which demonstrate the preference of BASHY for myelin-phagocytosing amoeboid cells, we next tested the probe in an in vivo mice model of MS, the EAE. EAE was chosen because it exhibits many similarities with clinical MS [40]. EAE-induced pathogenesis is characterized by an early acute phase of the disease associated with continuous aggravation of clinical symptoms of animal paralysis, which here translates into an increase in the clinical score values (0 to 5 grades), followed by a transient state of partial recovery. According to previous data, EAE animals develop an acute form of EAE (associated with higher clinical scores) around 17–18 days post-immunization (DPI), which is followed by a slight remission of symptoms around 21–23 DPI with partial motor recovery [41,42]. Therefore, in our experiment, mice were monitored for a period of 23 DPI to assess EAE-associated clinical signs. As a first and demonstrative assay to assess BASHY labeling of myelin debris after in vivo administration, and to enable the identification of myelin-rich cells within demyelinated lesions at peak and recovery phases, this probe was administered (100 µL (1 mM)/20 g body weight) to control animals and EAE-induced animals at 17 and 23 DPI (Figure 4A). To do so, we randomly divided our cohort into four groups: control animals receiving BASHY by retro-orbital intravenous injection (I.V.); control animals receiving BASHY by intraperitoneal injection (I.P.); EAE-induced mice receiving BASHY I.V.; and EAE-induced mice receiving BASHY I.P. (Figure 4A). In our first results, from flow cytometry analysis using live cells isolated from the cortex (CT), CB, and upper SC, we counted the number of activated macrophages/microglia (CD80^+^) (Appendix A) and the ones that were specifically stained with BASHY (CD80^+^/BASHY^+^). Indeed, we confirmed the presence of CD80^+^/BASHY^+^ in the CT, CB, and SC and observed that this number was increased in EAE-induced mice, more predominantly in the CB of EAE-induced animals following BASHY I.V. injection at 17 DPI, being over four times higher when compared to the control group (Figure 4B). Consistent with the flow cytometry analysis, our immunohistochemistry results showed the presence of lesioned areas (high nuclei area), characterized by the lack of myelin tracts (Mbp, red) and an extensive accumulation of cell infiltrates (DAPI, blue) in the cerebellar white matter of EAE animals. These lesions were mainly prevalent in EAE-induced mice after 17 DPI (Figure 4D and Appendix A) when compared to controls and EAE-induced animals after 23 DPI (Appendix A). Moreover, we evaluated EAE lesions at 17 and 23 DPI regarding microglia/macrophage accumulation (Iba1, red) and observed a greater accumulation of such Iba1-positive phagocytic cells in lesioned sites at 17 DPI (Figure 4C, Appendix A). Surprisingly, when assessing BASHY labeling, we could identify the presence of BASHY fluorescence within demyelinating lesioned areas at 17 DPI (Figure 4D and Supplemented Appendix A). Most importantly, even though we observed some non-specific binding of BASHY molecules to the granular layers of the cerebellum, the BASHY identification of lesion-associated microglia with excessive intracellular accumulation of lipids (foamy cells) was strongly corroborated by the co-localization of BASHY fluorescence and Iba1-positive cells, particularly evidenced for 17 DPI EAE in mice receiving I.V. injection (Figure 4C). 

## 4. Discussion

Myelin-associated alterations within demyelinated plaques are essential features of MS and related neurodegenerative disorders. Focusing on MS, myelin debris’ presence in such lesioned areas characterizes the newly formed active demyelinated plaques before they turn into inactive and silent sclerotic scars [43]. The inflow of microglia and peripheral macrophages is imperative to clear myelin debris and promote lesion recovery. That is why identifying phagocytosing cells and imaging the process of myelin clearance in vivo are fundamental to evaluate lesion progression/resolution and, therefore, accurately assess MS disease course. In the current study, we exploited a fluorescent molecule based on a boronic acid salicylidenehydrazone complex (BASHY), with a high affinity for lipid aggregates. Using BASHY, not only were we able to stain myelin debris and foamy cells effectively in ex vivo demyelinated samples, but we further identified myelin-phagocytosing cells in acute lesions of EAE-challenged mice. 

We began by analyzing BASHY fluorescent labeling after the demyelinating effect of LPC in the OCSCs. As pointed out already, following demyelination, myelin layers lose compactness and are converted into lipid-rich nonpolar fragments [31]. In fact, although we found BASHY-stained areas in non-induced OCSCs, likely resulting from mechanical injury during the tissue slicing procedure [44], we clearly observed a significant increase in fluorescent staining after induced demyelination, consistent with the primary accumulation of degenerated myelin after LPC induction. As the next step, to clarify BASHY discrimination between myelin structures, as well as to identify lipid-rich microglia within lesions, we stained for intact myelin, degraded myelin structures, and CNS glial cells (myelin-producing Ols, astrocytes, and microglia). Strengthening our hypothesis, we clearly show low BASHY co-localization with intact Mbp^+^ myelin sheaths and mature OLs. Instead, we see that our fluorescent probe stains a small percentage of initially formed non-compact Qd9-labeled structures but essentially co-localizes with Iba1-positive myelin-phagocytosing cells**,** thus emerging as a promising fluorescent dye for the brain imaging of amoeboid phagocytic microglia (foamy cells) based on its efficacy in labeling myelin debris. Apart from the available markers for intact myelin fibers, myelin debris at the final stage of myelin degradation has been commonly stained using Oil Red O dye [45], which is a non-specific marker that stains for general lipids [46], or tagged with pHRODO reagent to perform myelin debris engulfment assays [47]. A recent study also uses the lipophilic Nile Red dye as a marker to identify changes in the composition and/or polarity of myelin lipids in tissue sections following demyelination. However, final accumulation of myelin debris was still undetected [48]. Thus, our BASHY offers the possibility of discriminating detached myelin debris from intact or initially damaged myelin fibers, which, in turn, allows a more accurate visualization of myelin clearance by lesion-associated microglial cells in vitro and ex vivo with possible repurposing for in vivo models of demyelination.

Notably, partial labeling of astrocytes was also observed, which is in accordance with previous data evidencing the presence of myelin^+^ astrocytes in areas of active myelin breakdown [3,49,50], where they play a minor role in myelin clearance. However, their involvement is not as significant as that of microglia, also referred to as the professional phagocytes of the CNS and the main cells involved in myelin removal following demyelination [51]. Indeed, BASHY staining was mostly confined to microglial foamy cells and not only was the dye internalized, but BASHY could also accompany the myelin degradation pathway once it was found accumulated inside the lysosomes and stored into lipid bodies. Importantly, besides the observed BASHY labeling of microglia ex vivo, our results were even corroborated using primary cultures of microglia incubated with MD-BASHY. Here, we successfully showed MD-BASHY accumulation in microglia after in vivo phagocytosis. Altogether, these experiments confirm BASHY’s great stability after phagocytosis and internalization and, thus, corroborate its potential as an outstanding probe for the identification of microglia and myelin clearance in vivo. 

Microglia behave differently in physiological and non-physiological conditions or disease-associated environments. After LPC exposure, ramified microglia are almost entirely replaced by amoeboid cells, entering in an activated state [44,52]. Studies using OCSCs also described a shift in microglial phenotype after LPC induction, from an early accumulation of pro-inflammatory state cells toward anti-inflammatory microglia populations [35], which determines remyelination recovery. Whereas polarization into anti-inflammatory microglia are implicated in tissue regeneration through the secretion of growth and neurotrophic factors, depletion of such anti-inflammatory cells is strongly associated with impaired OL’s differentiation [53,54]. 

Consistent with the above findings, we clearly demonstrate a shift in cell morpho-logy toward the accumulation of amoeboid/activated cells after LPC exposure, but we also found our results indicative of a sequential change in phenotype over demyelination. At 18 h post-LPC, microglia are at a transient time point from an earlier homogeneous pro-inflammatory population that evolves into a mixed population co-expressing both anti- and pro-inflammatory markers (Arg1^+^ and iNOS^+^, respectively) until it finally forms a late anti-inflammatory microglial population. Interestingly, we noticed the presence of BASHY-stained myelin debris preferentially in amoeboid/activated cells, expressing either Arg1^+^/iNOS^-^ or Arg1^+^/iNOS^+^ at both time points (18 h and 48 h post-LPC). This is in accordance with studies from Miron and colleagues, revealing a phagocytic ability of both pro- and anti-inflammatory microglia, as phagocytic receptors increase in expression along the activation process of microglia [35]. As a result, we describe, for the first time, the use of a fluorescent probe for the imaging of myelin clearance by foamy microglia.

In the light of the former results, and based on BASHY ‘s affinity for foamy cells, we finally studied the potential of BASHY staining in vivo to identify demyelinated areas in a mouse model of MS, the EAE model. Studies using the EAE model confirm a decrease in Mbp staining in EAE mice [55], as well as the increased accumulation of myelin debris in such demyelinated lesions [56] during the acute phase of the disease, with a partially recovery until 23 DPI. Accordingly, in this work, we administered the BASHY dye through I.P. or retro-orbital I.V. injection, either at the disease’s peak (17 DPI) or later on during the recovery phase (23 DPI). Surprisingly, our data from the in vivo experiments demonstrated that BASHY could reach the CNS, as we successfully isolated BASHY^+^ activated microglia from CB, CT, and upper SC, with an increase in BASHY^+^/CD80^+^ cells in the CB of EAE-challenged mice receiving the I.V. injection at 17 DPI. It is already known that EAE induction in C57BL/6 mice is characterized by multifocal areas of demyelination mainly in the CB and SC [39]. In the SC, inflammatory lesions in particular are initially formed at the lower level of the lumber cord before spreading to the upper level of the SC [57], which might explain the evident augmentation in the number of BASHY^+^/CD80^+^ cells in the CB when compared to that found in the upper SC at disease peak in mice receiving I.V. injection. Moreover, in tissue sections from EAE-challenged mice, we observed the formation of early lesioned areas in the white matter of the cerebellum at 17 DPI that seemed slightly less evident at 23 DPI and absent in controls, confirming the efficacy of the EAE model. More importantly, aside from the non-specific binding of BASHY, we detected the co-localization of the BASHY fluorescent signal with foamy microglia in areas of active myelin disruption, particularly evident in EAE-challenged mice receiving I.V. injection at 17 DPI. 

We propose that the greater efficacy of BASHY staining in mice receiving I.V. injection is due to the fact that intravenously administered BASHY is more quickly and easily delivered in higher concentrations into the CNS, whereas intraperitoneally, BASHY is expected to be absorbed and metabolized in other organs, hence reaching CNS at a much slower rate and reduced concentration [58]. However, the evident absence of the BASHY signal in the white matter of control OCSCs, as well as the decrease in staining in tissue sections from EAE mice after 23 DPI, sustains our hypothesis that BASHY not only enters the brain but also accumulates within active lesioned areas specifically enriched in myelin-phagocytosing cells. Nonetheless, additional experiments are already underway to support these first in vivo results. These include the optimization of BASHY (increasing stability and selectivity), which is to be tested using different demyelinating models and time points, and the evaluation of alternative administration routes, levels of toxicity, and absorption rate after in vivo uptake.

## 5. Conclusions

In conclusion, with this study, we aimed to respond to the urgent need for more specific imaging techniques for the study, clinical diagnosis, and monitoring of MS. In light of this, we utilize a fluorescent molecule with great specificity for myelin debris and, therefore, capable of identifying myelin-phagocytosing cells, which furnished highly promising results in vivo. Indeed, we are convinced that BASHY offers the unique possibility to be used non-invasively to identify MS demyelinating lesions, as indicators of disease stage and progression, in longitudinal pre-clinical studies using live animals, but we are hoping this evolves into the clinical monitoring of MS patients.

## Figures and Tables

**Figure 1 cells-10-03163-f001:**
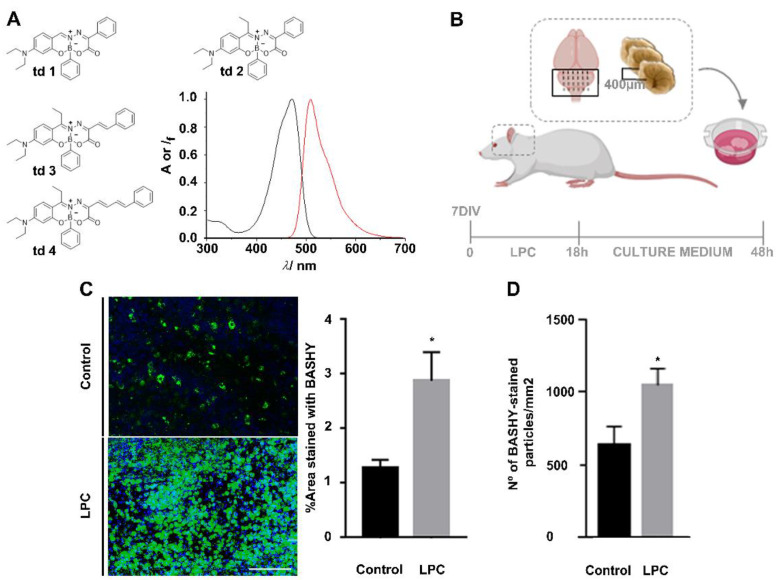
BASHY labels myelin debris. (**A**) Structures of BASHY candidates td**1–4** to stain myelin fragments and absorption (black line) and emission (red line) fluorescence spectra of dye 2 in toluene as apolar solvent. (**B**) Scheme of demyelinating ex vivo model of organotypic cerebellar slice cultures (OCSCs). OCSCs from 10 postnatal day (P10) rats were incubated with lysophosphatidylcholine (LPC) at 7 days in vitro (DIV) for 18 h to induce demyelination. After LPC exposure, OCSCs were incubated with new neurobasal fresh culture medium for a recovering period of 30 h. (**C**) Representative images and respective quantitative analysis of the percentage of area stained with BASHY td**2** (green) in control and demyelinated OCSCs, 48 h after LPC induction, and imaged in the cerebellar white matter. * *p* < 0.05 vs. control (*t*-test). (**D**) Quantitative analysis of the number of BASHY-stained particles (green) in control and demyelinated OCSCs, 48 h after LPC induction. * *p* < 0.05 vs. control (*t*-test). Data are representative of four independent experiments. A total of 20 z-planes were analyzed per field. Scale bar equals 100 microns (created with Biorender.com).

**Figure 2 cells-10-03163-f002:**
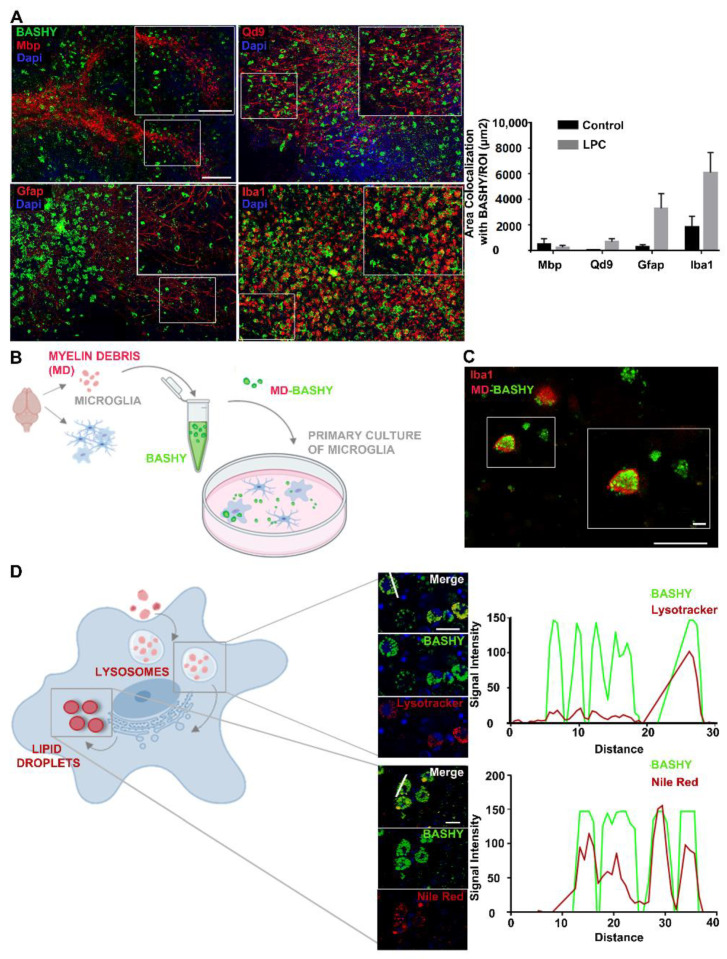
BASHY targets myelin-phagocytosing macrophages/microglia and images of myelin intracellular degradation pathway. (**A**) Representative images from the cerebellar white matter of organotypic cerebellar slice cultures (OCSCs) and respective quantitative analysis of BASHY (green) co-localization with mature oligodendrocytes/intact myelin (Mbp, red), degenerated myelin (Qd9, red), astrocytes (Gfap, red), and microglia/macrophages (Iba1, red) 48 h post-LPC induction (magnification 20X). Scale bar equals 120 microns. Results are mean ± SEM. Data are representative of four independent experiments. In total, 17 to 25 z-planes were analyzed per field. (**B**) Schematic representation of BASHY staining of myelin debris and posterior incubation in primary cultured microglia. Myelin debris was collected from P10 rats and stained with BASHY probe. At the same time, microglia were isolated from mixed glial cultures prepared from P10 rats and cultured for another 24 h, after which, they were incubated with the previously stained BASHY-labeled myelin debris (MD-BASHY). (**C**) Representative images of MD-BASHY (green) in culture (scale bar equals 120 microns) and internalized by microglia (Iba1, red) (scale bar equals 20 microns). (**D**) After internalization, excessive myelin-derived cholesterol aggregates accumulate inside lysosomes before being transported to the endoplasmic reticulum and finally stored into lipid droplets. Schematic figure of myelin intracellular degradation pathway and representative images of BASHY (green) co-localization with lysosomes (Lysotracker, red; upper panel) and lipid droplets (Nile Red, red; lower panel) with specific fluorescence signal intensity profiles of the identified region (white line). Scale bar equals 20 microns. Gfap, glial fibrillar acidic protein; Mbp, myelin basic protein; Iba1, ionized calcium binding adaptor molecule 1 (created with Biorender.com).

**Figure 3 cells-10-03163-f003:**
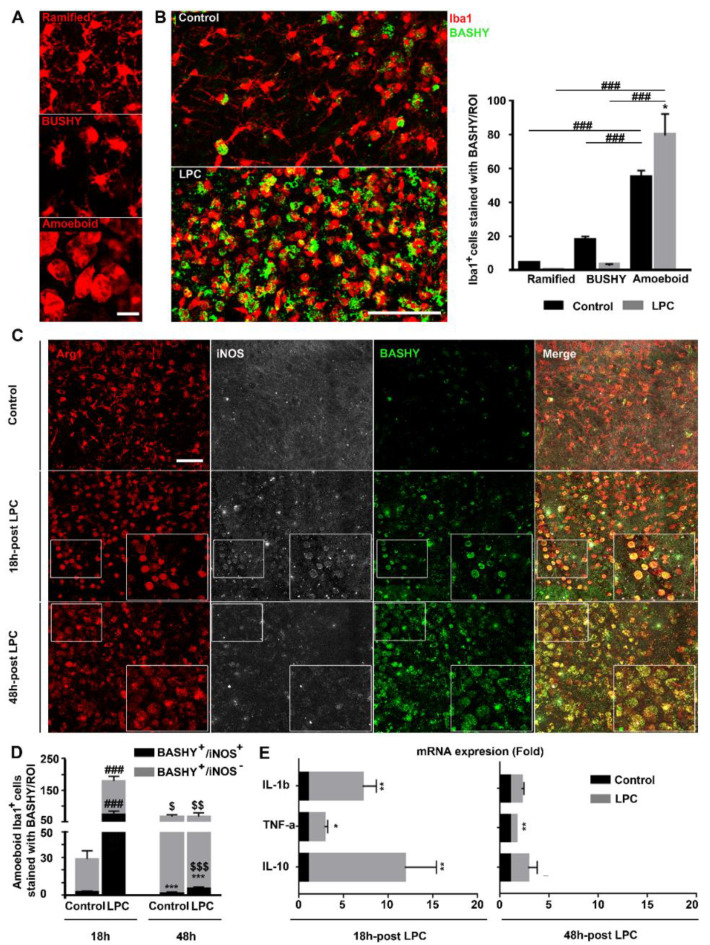
BASHY selectively targets amoeboid phagocytic Iba1^+^ cells after demyelination. (**A**) Representative images of differential macrophage/microglial (Iba1, red) morphologies, including ramified, bushy, and amoeboid morphologies, imaged in the cerebellar white matter. Scale bar equals 20 microns. (**B**) Representative images and respective quantitative analysis of organotypic cerebellar slice cultures (OCSCs), immunostained for microglia/macrophages (Iba1, red) and further labeled with BASHY dye (BASHY, green) to observe cell morphological changes over demyelination and BASHY preference over cell morphologies, 48 h post-incubation with LPC. Scale bar equals 100 microns. Results are mean ± SEM. * *p* < 0.05 vs. respective control. ^###^
*p* < 0.001 for amoeboid LPC vs. bushy LPC; for amoeboid LPC vs. ramified LPC; for amoeboid control vs. bushy control; and for amoeboid control vs. ramified control. Data are representative of two independent experiments. (**C**) Representative images from the cerebellar white matter of control and LPC-induced OCSCs at 18 h and 48 h post-induction that were immunostained to observe microglial anti-inflammatory marker Arginase (Arg1, red) and inducible nitric oxide synthase, a common marker of inflammatory microglia (iNOS, white), and to further assess co-localization with BASHY molecule (BASHY, green). Scale bar equals 120 microns. (**D**) Quantitative analysis of BASHY co-localization with anti-inflammatory (iNOS-) and pro-inflammatory (iNOS^+^) microglia at each time point. Results are mean ± SEM. *** *p* < 0.001 for BASHY^+^/iNOS^+^ vs. respective BASHY^+^/iNOS^-^; ^###^
*p*<0.001 for BASHY^+^/iNOS^+^ LPC vs. respective control; and for BASHY^+^/iNOS^-^ LPC vs. respective control. ^$^
*p* < 0.05 for BASHY^+^/iNOS^-^ control 48 h vs. respective 18 h; ^$$^ *p* < 0.01 for BASHY^+^/iNOS^+^ LPC 48 h vs. respective 18 h; and ^$$$^
*p* < 0.001 for BASHY^+^/iNOS^+^ LPC 48 h vs. respective 18 h. Data are representative of two independent experiments. (**E**) mRNA expression of pro-inflammatory (IL-1β and TNF-α) and anti-inflammatory (IL-10) cytokines in control and LPC-induced OCSCs at each time point. Results are expressed as mean ± SEM. * *p* < 0.05, ** *p* < 0.01 vs. respective control.

**Figure 4 cells-10-03163-f004:**
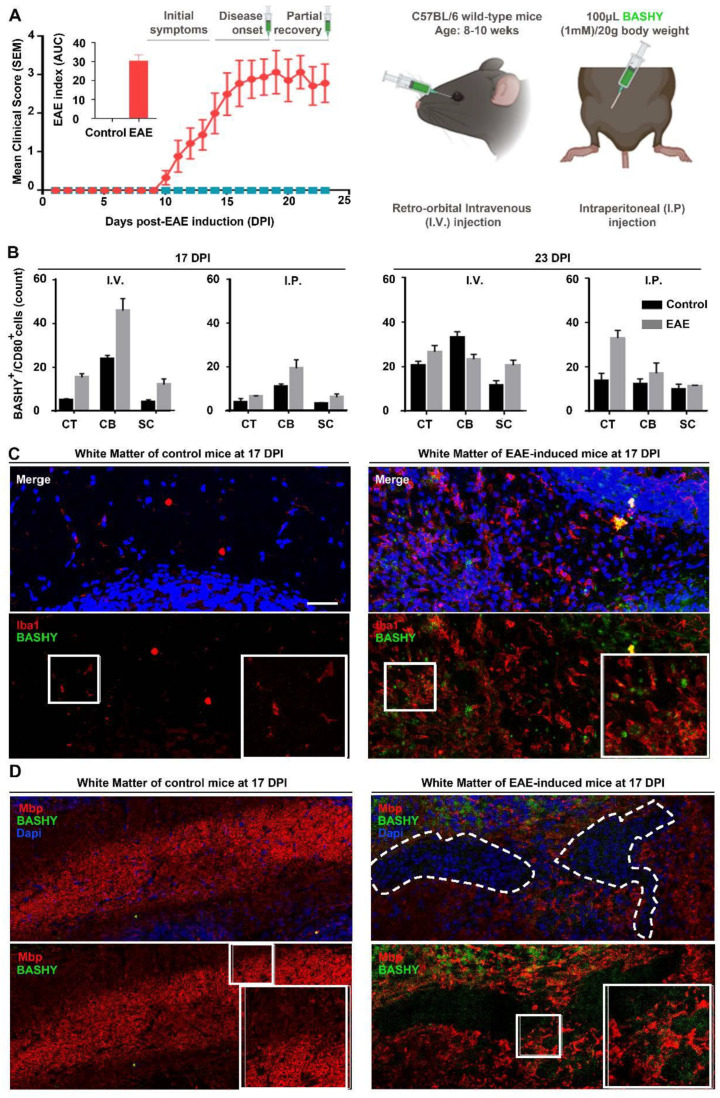
BASHY labels lesion-associated myelin-phagocytosing microglia in EAE-induced mice. (**A**) Clinical observations of control and EAE-challenged mice during 23 days after immunization, and global scheme of BASHY administration in vivo. Following EAE induction, mice clinical symptoms are characterized by an increase in paralysis beginning at the tail until animals reach quadriplegia and death. The clinical score was given during each day of the experiment following a 5-point standardized scale, and the area under the curve (AUC) of the overall disease severity was calculated for each mouse. A total of 100 µL of BASHY molecule was administered (1 mM/20 g body weight) at both 17 (disease peak) and 23 DPI either intravenously (I.V) or intraperitoneally (I.P). (**B**) Flow cytometry analysis was performed using cells from cortex (CT), cerebellum (CB), and spinal cord (SC) from control and EAE-induced mice injected with BASHY. Cells were probed with the antibody CD80 to stain for activated microglia. Panels (**C**) and (**D**) are representative images of brain sections from control and EAE-induced mice 17 DPI, injected with BASHY I.V. Brain sections were stained for microglia/macrophages (Iba1, red) in (**C**) or stained for cell nuclei (DAPI, blue) and compact myelin sheaths (myelin basic protein, Mbp, red) in (**D**) and imaged in the cerebellar white matter. Magnification 20X. Scale bar equals 200 µm (created with Biorender.com).

## Data Availability

The data presented in this study are available on request from the corresponding author.

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
