# Peer review of "BASHY Dye Platform Enables the Fluorescence Bioimaging of Myelin Debris Phagocytosis by Microglia during Demyelination"

_cells, 2021, doi:10.3390/cells10113163_

Round 1

Reviewer 1 Report

I enjoyed reading the manuscript that addressed a key technique in investigating demyelination and debris clearance. While manuscript is well written, a careful revision is required as well as authors may address some other issues as follows.  

  • Introduction is quite well written but a sudden jump to EAE before describing anything is found. Authors are suggested to make it clear
  • Authors should revise the whole manuscript to avoid small mistakes like this (104-μm) and use full form followed by abbreviation
  • Author may double check this (myelin binding protein (MBP)
  • Figure 1: I assume authors measured intensity using Image J to get the result. Authors are suggested to provide details how Image J is used to measure intensity. Just curious, if it is looked closely the image shows there are myelin debris in the Control, then question is demyelination occurs without LPC challenge. Second, the bar graph says the ratio 1: 2; means 2-fold increase the intensity but image is showing there is more myelin debris in the LPC challenge image. Authors can try to measure particle/debris by setting ImageJ the way cells are measured.  Figure 1B and other schematic is possibly drawn by Biorender or other tools. Authors can acknowledge them.
  • Authors can revise the manuscript to make gene IDs homogenous. Ideally, gene ids are written as Gfap or Iba1, Mbp. Oligodendrocyte and oligodendrocyte, mature OLs…. both were found. In multiple places authors used abbreviations in other places not for may cells or regions. Authors should revise throughout the manuscript carefully  
  • Authors could change the word from treatment to induced or administrated as it is not treating any disease (LPC-treatment) throughout
  • BASHY staining identify demyelination. There is another recent work showing Nile Red can show very subtle change in cuprizone mode (PMID: 33593907). Authors could discuss what can detect better. Although used Nile red in their study. Another suggestion, I think for myelin debris, phagocytosis study Cuprizone model is better as it gives massive demyelination and glial activation. Here one recent example PMID: 33593907. May be author could think to do study in the future.
  • Authors could bring couple of images from Supplementary to the main manuscript such as Figure 4. It might help reader not to go and back to the manuscript. 

Author Response

Cells Reviewers comments

We thank the reviewers for their helpful comments. We reviewed the manuscript to address the following issues, added more data and we feel that the manuscript is now greatly improved.

Reviewer 1

I enjoyed reading the manuscript that addressed a key technique in investigating demyelination and debris clearance. While manuscript is well written, a careful revision is required as well as authors may address some other issues as follows.

We thank the Reviewer for the valuable comments, a detailed answer to each point is given:

  • Introduction is quite well written but a sudden jump to EAE before describing anything is found. Authors are suggested to make it clear

As suggested, we modified the introduction accordingly by adding additional information about the relevance of the EAE model to study in vivo demyelination-associated pathology, Line 74

  • Authors should revise the whole manuscript to avoid small mistakes like this (104-μm) and use full form followed by abbreviation

The whole manuscript was revised and identified mistakes were corrected.

  • Author may double check this (myelin binding protein (MBP)

The incorrection was changed to myelin basic protein, Line 230.

  • Figure 1: I assume authors measured intensity using Image J to get the result. Authors are suggested to provide details how Image J is used to measure intensity. Just curious, if it is looked closely the image shows there are myelin debris in the Control, then question is demyelination occurs without LPC challenge. Second, the bar graph says the ratio 1: 2; means 2-fold increase the intensity but image is showing there is more myelin debris in the LPC challenge image. Authors can try to measure particle/debris by setting ImageJ the way cells are measured.  Figure 1B and other schematic is possibly drawn by Biorender or other tools. Authors can acknowledge them.

In figure 1, we calculated the % of area stained with BASHY. As suggested, and to avoid misunderstandings we explained in the methods section that we used a cut-off intensity threshold value for each condition, which corresponds to a minimum intensity due to specific staining above background values. Then, automatically we measured the % of stained area using Image J. Line 192

Regarding the observed presence of myelin debris in control OCSC, we clarified in the text (Discussion section) that this can be observed as a result of the mechanical injury during tissue slicing procedure. Line 602

To better corroborate our results, as suggested, we measured particle/debris automatically using ImageJ as described in Methods (Line 196) and the results were described (Line 373) and added to Figure 1.

Finally, we also recognized and acknowledged Biorender Line 743 and in each figure legend.

  • Authors can revise the manuscript to make gene IDs homogenous. Ideally, gene ids are written as Gfap or Iba1, Mbp. Oligodendrocyte and oligodendrocyte, mature OLs…. both were found. In multiple places authors used abbreviations in other places not for may cells or regions. Authors should revise throughout the manuscript carefully

The whole manuscript was revised to rectify all gene IDs, as suggested.

  • Authors could change the word from treatment to induced or administrated as it is not treating any disease (LPC-treatment) throughout

We revised the whole manuscript and changed from LPC-treated to LPC-induction/LPC exposure, as suggested.

  • BASHY staining identify demyelination. There is another recent work showing Nile Red can show very subtle change in cuprizone mode (PMID: 33593907). Authors could discuss what can detect better. Although used Nile red in their study. Another suggestion, I think for myelin debris, phagocytosis study Cuprizone model is better as it gives massive demyelination and glial activation. Here one recent example PMID: 33593907. May be author could think to do study in the future.

As suggested, we discussed the advantages of BASHY to label debris, when compared with the most commonly used lipid markers. Line 615. Also, our results using the EAE model are intended to deliver proof-of-principle. In future experiments, the dye will be tested using other demyelinated models as the Cuprizone one.

  • Authors could bring couple of images from Supplementary to the main manuscript such as Figure 4. It might help reader not to go and back to the manuscript. 

As suggested, we brought Supplementary Fig 4A to the main manuscript to allow side by side visualization of the results regarding the demyelination, debris accumulation and BASHY staining in control and EAE-induced mice at the peak of the disease after I.V injection. Line 572

Reviewer 2 Report

In the manuscript by Pinto et al. authors developed a new BASHY fluorophore and seek to establish it as a novel marker that binds myelin debris and thus allows to image myelin clearance and engulfment by phagocytes. Authors use rat ex vivo organotypic cerebellar brain slices and the in vivo EAE mouse model of demyelination to show increased BASHY signals and more pronounced colocalization with phagocytic microglia after demyelination. The development of novel imaging tools to better assess myelin engulfment is of high interest to the field; therefore, enthusiasm for the study is high. However, currently, some major conclusions in the manuscript are not sufficiently supported by the provided data. These and other concerns that should be addressed before publication are outlined below:

Major comments:

  • Authors hypothesize that BASHY binds primarily non-polar myelin debris and not intact myelin. Given that most myelin in control animals is intact, it is unclear why authors detect pronounced and widespread BASHY signals (for all four compounds) in sections from control animals in Fig. 1 and Suppl Figs. 1-2? These data suggest that BASHY labels intact myelin, plasma membranes, or other cellular material in control tissue as well. BASHY labeling also shows a similar pattern in control and LPC-treated animals (although more abundant in LPC-treated animals), further supporting that many of the BASHY signals in LPC-treated animals are associated with other cellular material than myelin debris. It is unclear how authors distinguish between BASHY-labeled myelin debris and other cellular material, including intact myelin and plasma membranes in LPC-treated animals? Given that the authors suggest that BASHY can be used as a marker of myelin debris engulfment, it is, however, important to prove that BASHY allows for the distinction between myelin debris and intact myelin or other cellular materials.
  • It is standard to run a quality control for colocalization analysis that shows that colocalization is not just by chance. This is particularly important because many of the BASHY signals shown in Fig. 2A are not colocalized with myelin debris. The authors could rotate one image channel by 180 degrees and run the same colocalization analysis to test for colocalization by chance.
  • Authors describe colocalization of BASHY with GFAP and note that this colocalization is less pronounced compared to microglia/macrophages. Therefore, the authors conclude that BASHY fluorescence is strongly confined to phagocytic microglia/macrophages. However, it looks like the number of BASHY signals increases significantly in GFAP+ astrocytes in LPC-treated animals compared to controls (Fig. 2A), suggesting that BASHY is increasingly labeling astrocytes following demyelination. The authors will need to perform additional experiments to establish that BASHY specifically labels phagocytic microglia/macrophages. This could include feeding astrocyte cultures with BASHY-labeled myelin debris.
  • In the images in Fig. 2D, the authors use maximum projected Z stacks (line 168) to make conclusions regarding BASHY colocalization with lysosomes and lipid droplets. As maximum intensity projections merge fluorescence from all the planes of a section, the observed colocalized fluorescence may or may not be in the same plane. Authors should analyze sub-stacks or 3D projections to ensure colocalization in the z-plane. Representative images should also be 3-D renderings or at least show orthogonal views of all channels for all z-planes. The authors should also state how many z-planes were taken and analyzed per field of view.
  • There is ample published work that identified the spinal cord as one of the earliest and most severely affected CNS regions in the EAE mouse model. Accordingly, demyelination in the spinal cord is typically more pronounced compared to other regions, particularly grey matter areas such as the cortex. Therefore, it is surprising that the number of BASHY+/CD80+ cells in the cortex was comparable or even higher than the number in the spinal cord at both time points and using both injection paradigms (Fig. 4B). This raises concerns about the validity of the provided analysis. Author will need to prove that demyelination shows the expected pattern and clarify this discrepancy? Related to this point, it is also unclear why the FACS counts of BASHY+/CD80+ are so low (ranging from 10-50 cells per CNS region) when a single region contains thousands of microglia/macrophages. This data is also not convincing and warrants further validation.
  • Representative images shown in Fig. 2-4 (and the corresponding Suppl. Figs.) appear to show different subregions of the cerebellum. Images showing more similar staining patterns for DAPI/cellular markers should be provided from comparable subregions. If quantification were performed on different cerebellar subregions, authors would also need to repeat these analyses using more comparable regions of interest.
  • Under healthy conditions, the vast majority of microglia in the brain typically show a ramified morphology. Therefore, it is surprising that the authors found almost 55% of microglia having an ameboid morphology in control animals (Fig. S3A). How do authors explain this unexpected finding?
  • Authors used Triton X-100, whose monomers insert into cell membranes and destruct the compactness and integrity of the lipid bilayer, to stain tissue sections with antibodies against cell-specific proteins. On these same slides, the authors used BASHY dye to label myelin debris. Does Triton X-100 interaction with the membrane have an effect on BASHY labeling of myelin? Have authors tested to use BASHY on sections stained without using Triton X-100?
  • Authors should discuss their findings and the advantages and disadvantages of the BASHY dye with respect to existing imaging tools for myelin engulfment in the field, for instance, pH-Rodo labeling of myelin debris.
  • The number of biological replicates and the number of cells/fields of view analyzed per animal should be stated for each experiment in the figure legends.

Minor Comments:

  • DAPI fluorescence (or a similar landmark marker) should be used in Figs. 1 and S1 to visualize that BASHY signals in comparable regions of the cerebellar slices are shown across conditions.
  • In lines 309-310, the authors describe more marked emission intensity for samples subjected to LPC compared to controls. The provided images all appear to show similar intensities, though. It would be helpful to quantify the intensity data in Fig. S1 to better support this point.
  • To better highlight the shifts in spectra following chemical modification, absorption and emission curves should also be shown for dyes 1, 3, and 4 in Fig. 1. Likewise, authors should also show red channel images for dyes 1 and 2 in Fig. S1 to better support that there is no bleed-through signal in the red channel of these dyes.
  • The GFAP staining in Fig. 2 looks rather dim and appears of low quality as the typical fibrillary GFAP signal cannot be appreciated. Authors may need to optimize staining first to be able to perform meaningful colocalization analysis.
  • The description and organization of the results in Fig. 3 are hard to follow. The manuscript would largely benefit from a clearer presentation.
  • The authors need to provide the clinical scores of EAE mice in Fig. 4.
  • The IHC data in Fig. 4C should be quantified.

Author Response

Reviewer 2

In the manuscript by Pinto et al. authors developed a new BASHY fluorophore and seek to establish it as a novel marker that binds myelin debris and thus allows to image myelin clearance and engulfment by phagocytes. Authors use rat ex vivo organotypic cerebellar brain slices and the in vivo EAE mouse model of demyelination to show increased BASHY signals and more pronounced colocalization with phagocytic microglia after demyelination. The development of novel imaging tools to better assess myelin engulfment is of high interest to the field; therefore, enthusiasm for the study is high. However, currently, some major conclusions in the manuscript are not sufficiently supported by the provided data. These and other concerns that should be addressed before publication are outlined below:

We thank again Reviewer 2 for the time spent revising the manuscript and, once again, for the valuable comments. Answers to Reviewer’s comments are below:

Major comments:

  • Authors hypothesize that BASHY binds primarily non-polar myelin debris and not intact myelin. Given that most myelin in control animals is intact, it is unclear why authors detect pronounced and widespread BASHY signals (for all four compounds) in sections from control animals in Fig. 1 and Suppl Figs. 1-2? These data suggest that BASHY labels intact myelin, plasma membranes, or other cellular material in control tissue as well. BASHY labeling also shows a similar pattern in control and LPC-treated animals (although more abundant in LPC-treated animals), further supporting that many of the BASHY signals in LPC-treated animals are associated with other cellular material than myelin debris. It is unclear how authors distinguish between BASHY-labeled myelin debris and other cellular material, including intact myelin and plasma membranes in LPC-treated animals? Given that the authors suggest that BASHY can be used as a marker of myelin debris engulfment, it is, however, important to prove that BASHY allows for the distinction between myelin debris and intact myelin or other cellular materials.

To justify our results showing BASHY stained areas in control OCSC, we added in the discussion part (Line 602) that it is expected, to some extent, myelin destruction and glial reactivity in control OCSC as a result of the mechanical injury during the slicing procedure. Regarding BASHY labeling of intact myelin and plasmatic membranes, we only observed slight co-localization with Mbp in figure 2B, which means that BASHY does not have much affinity for intact myelin structures. Instead, it seems that it stains early Qd9-positive degraded myelin vesicles. Finally, as stated in the Introduction, these BASHY complexes already proved the ability to discriminate between lipid vesicles and plasmatic membranes as reported in previous studies as the one from reference [19], doi:10.1002/chem.201503943.

  • It is standard to run a quality control for colocalization analysis that shows that colocalization is not just by chance. This is particularly important because many of the BASHY signals shown in Fig. 2A are not colocalized with myelin debris. The authors could rotate one image channel by 180 degrees and run the same colocalization analysis to test for colocalization by chance.

We understand the Reviewer’s concern and we better described in the Methods (Line 243) how we measured the co-localization of BASHY with each specific marker. Indeed, since we did it for each specific Z stack we are convinced that indeed we are observing co-localization and these resulted are further corroborated by the 3D images added in Supplementary Figure 2H for colocalization with Nile-Red and/or Lysotracker. Nevertheless, we performed the suggested rotation and new analysis and we obtained exactly the same results, so we decided not to add those to the present manuscript.

  • Authors describe colocalization of BASHY with GFAP and note that this colocalization is less pronounced compared to microglia/macrophages. Therefore, the authors conclude that BASHY fluorescence is strongly confined to phagocytic microglia/macrophages. However, it looks like the number of BASHY signals increases significantly in GFAP+ astrocytes in LPC-treated animals compared to controls (Fig. 2A), suggesting that BASHY is increasingly labeling astrocytes following demyelination. The authors will need to perform additional experiments to establish that BASHY specifically labels phagocytic microglia/macrophages. This could include feeding astrocyte cultures with BASHY-labeled myelin debris.

As suggested, we performed additional experiments incubating, in separate microglia and astrocytes with isolated BASHY-stained myelin debris (MD-BASHY). By flow cytometry analysis using the Cytek® Aurora flow cytometer we compared the capacity of microglia and astrocytes to phagocytose myelin debris. The new results described (Line 402) and added to the Supplementary Fig 2C-G clearly show that microglia have a higher affinity to phagocyte MD-BASHY, supporting our previous interpretation from the OCSC.

  • In the images in Fig. 2D, the authors use maximum projected Z stacks (line 168) to make conclusions regarding BASHY colocalization with lysosomes and lipid droplets. As maximum intensity projections merge fluorescence from all the planes of a section, the observed colocalized fluorescence may or may not be in the same plane. Authors should analyze sub-stacks or 3D projections to ensure colocalization in the z-plane. Representative images should also be 3-D renderings or at least show orthogonal views of all channels for all z-planes. The authors should also state how many z-planes were taken and analyzed per field of view.

As suggested, we re-measured BAHSY co-localization with lysosomes and Nile Red from sub-stacks, instead of evaluating maximum projected Z stacks, to make sure the labeling is in the same plane. The new data and representative images were added to Figure 2D (Line 431), and methodology used was clarified (Line 198). Additional 3D images were added to Supplementary Figure 2H to corroborate the co-localization results.

  • There is ample published work that identified the spinal cord as one of the earliest and most severely affected CNS regions in the EAE mouse model. Accordingly, demyelination in the spinal cord is typically more pronounced compared to other regions, particularly grey matter areas such as the cortex. Therefore, it is surprising that the number of BASHY+/CD80+ cells in the cortex was comparable or even higher than the number in the spinal cord at both time points and using both injection paradigms (Fig. 4B). This raises concerns about the validity of the provided analysis. Author will need to prove that demyelination shows the expected pattern and clarify this discrepancy? Related to this point, it is also unclear why the FACS counts of BASHY+/CD80+ are so low (ranging from 10-50 cells per CNS region) when a single region contains thousands of microglia/macrophages. This data is also not convincing and warrants further validation.

We agree with Reviewer that upon EAE induction the spinal cord is one of the most affected regions and understand the surprise about the low number of BASHY+/CD80+. However, we may add that microglia only accounts for 5-10% of the total glial cell population (https://doi.org/10.1016/j.tice.2020.101438) and although it would markedly increase upon demyelination, since we only analyzed 30 000 events on the flow cytometer for the EAE-animal samples (information added now in the Methods Line 317) and from those we observed a marked increase in CD80+ microglia following EAE induction, as now shown in Supplementary Figure 4A. From those a much lower number was positive for BASHY, as shown in Figure 4B, reason why we still need to perform further improvements on BASHY in vivo stability. These studies are under way and will be included in follow-up work.

  • Representative images shown in Fig. 2-4 (and the corresponding Suppl. Figs.) appear to show different subregions of the cerebellum. Images showing more similar staining patterns for DAPI/cellular markers should be provided from comparable subregions. If quantification were performed on different cerebellar subregions, authors would also need to repeat these analyses using more comparable regions of interest.

All images were taken from the white matter region of the cerebellum. DAPI staining was added to all images, except the ones in Figure 2 and Supplementary Figure 1. In Figure 2, we used a secondary antibody emitting in the blue wavelength, and in Supplementary Figure 1C, images were only taken to visualize differences in the fluorescence capacity of the different probes, so DAPI was not used in this first experiment. Even so, this observation will be considered in future experiments.

  • Under healthy conditions, the vast majority of microglia in the brain typically show a ramified morphology. Therefore, it is surprising that the authors found almost 55% of microglia having an ameboid morphology in control animals (Fig. S3A). How do authors explain this unexpected finding?

As explained above, it is expected, to some extent, myelin destruction and microglia activation in control OCSC as a result of the mechanical injury during the slicing procedure that was discussed Line 602. As so, this may explain the number of amoeboid cells found in control conditions. Nevertheless, LPC induction greatly increased the number of amoeboid Iba+ cells (Supplementary Figure 3A), as well as the number of amoeboid Iba+/BASHY+ cells (Figure 3B).

  • Authors used Triton X-100, whose monomers insert into cell membranes and destruct the compactness and integrity of the lipid bilayer, to stain tissue sections with antibodies against cell-specific proteins. On these same slides, the authors used BASHY dye to label myelin debris. Does Triton X-100 interaction with the membrane have an effect on BASHY labeling of myelin? Have authors tested to use BASHY on sections stained without using Triton X-100?

In the experiment where we evaluated BASHY co-localization with Nile Red and Lysotracker, OCSC were not submitted to any treatment with Triton-X. As so, we have not seen any differences in BASHY staining either in the presence of absence of Triton X-100 use.

  • Authors should discuss their findings and the advantages and disadvantages of the BASHY dye with respect to existing imaging tools for myelin engulfment in the field, for instance, pH-Rodo labeling of myelin debris.

As suggested, we discussed the advantages of BASHY to label debris, when compared with the most commonly used lipid markers. Line 615

  • The number of biological replicates and the number of cells/fields of view analyzed per animal should be stated for each experiment in the figure legends.

The information regarding the number of replicates and fields of view analyzed per animal was added to the respective Figure legends, as suggested.

Minor Comments:

  • DAPI fluorescence (or a similar landmark marker) should be used in Figs. 1 and S1 to visualize that BASHY signals in comparable regions of the cerebellar slices are shown across conditions.

All images are from the cerebellar White Matter region. DAPI staining was added to images in Figure 1. In Supplementary Figure 1, images were only representative to visualize differences in the fluorescence capacity of the different probes, so DAPI was not used in this first experiment. However, this observation will be considered in future experiments.

  • In lines 309-310, the authors describe more marked emission intensity for samples subjected to LPC compared to controls. The provided images all appear to show similar intensities, though. It would be helpful to quantify the intensity data in Fig. S1 to better support this point.

We only quantified the % of td2 (BASHY) stained area in control and LPC-induced slices (Figure 1C) because of its superior performance as we explain in Line 357. To avoid misunderstandings, we removed the statement “more marked emission intensity for samples subjected to LPC” Line 371.

  • To better highlight the shifts in spectra following chemical modification, absorption and emission curves should also be shown for dyes 1, 3, and 4 in Fig. 1. Likewise, authors should also show red channel images for dyes 1 and 2 in Fig. S1 to better support that there is no bleed-through signal in the red channel of these dyes.

As suggested, we added the spectra for dyes 1 and 3 to Supplementary Figure_1.

Regarding the presence of the red channel: we only looked for the red shifting when analyzing probes 3 and 4 because the observed red-shift of the absorption and fluorescence spectra on going from 1 and 2 to 3 or 4 result from the increased pi-conjugation length in the salicylidenehydrazone backbone, thereby lowering the HOMO-LUMO gap of the dyes (DOI: 10.1002/chem.202001623). This behavior is typically observed for cyanine-like dyes.

  • The GFAP staining in Fig. 2 looks rather dim and appears of low quality as the typical fibrillary GFAP signal cannot be appreciated. Authors may need to optimize staining first to be able to perform meaningful colocalization analysis.

To avoid misunderstandings about staining quality and co-localizations, we increased the quality and zoomed in the images in Figure 2A and Supplementary Figure_2A.

  • The description and organization of the results in Fig. 3 are hard to follow. The manuscript would largely benefit from a clearer presentation.

To clarify Figure 3 presentation, we edited the data indicated in 3B, D and E.

  • The authors need to provide the clinical scores of EAE mice in Fig. 4.

As suggested, instead of providing a scheme we presented the actual information on the severity of the clinical EAE phenotype (clinical score and AUC) in Figure 4A.

  • The IHC data in Fig. 4C should be quantified

Our results using the EAE model are only demonstrative as we are still optimizing our BASHY in vivo stability as indicated above. In future experiments, the improved dye will be certainly tested in a higher number of animals using other demyelinated models as the Cuprizone one and results will be included in a future publication.

Reviewer 3 Report

Pinto and colleagues investigate the potential of four green-emitting fluorophores that recognize myelin debris in vitro and in vivo neuroinflammatory milieu. They provide extensive supplementary data on how these dyes were developed accompanied with structural characterization. The one that yielded optimal results (BASHY molecule), was further applied to primary microglia cultures and injected to MOG EAE-challenged mice.

The following comments should be taken into consideration:

Abstract

In its current form, it fits more to a review rather than a research article. As clearly stated in the instructions to authors, it should follow the style of structured abstracts, but without headings (i.e background, methods, results and conclusion). Authors are advised to stick to the aforementioned rules and reform it.

Introduction

Lines 34-39: Effective remyelination does not depends solely on removal of myelin debris by microglia. Oligodendrocyte apoptosis and endogenous neural precursor cell mobilization and proliferation should also be taken into account.

The last 2 paragraphs could be merged into one, describing concisely the purpose of the study.

Line 70: EAE-challenged not EAE-treated mice

Materials and Methods

Line 162 & 164: Nile Red & Filipin - which source/company?

Line 168: replace "ImageJ (Fiji Is Just)" with "ImageJ" or "Fiji (Fiji Is Just ImageJ)"

Results

The results section needs major revisions. First of all, numerical data must be accurately presented (e.g cells/mm2) because the approximate percentages provided (~90%, ~30% , ~6% etc.) currently denigrate the scientific credibility and render this section an extension of a discussion. Additionally, one sentence per result section in the end of the corresponding paragraph, is enough to conclude the findings. All the remainder discursive information, should be integrated into discussion interpreting the findings.

Probe candidates 1-4 should be alternatively referred to both text and Figure 1 as e.g test dye 1 (td1), td2 instead of just 1, 2

Line 346: This is particularly evident (in or) from... not evidenced by

Line 397: "foamy-like microglia" is not commonly used as foamy macrophages. Maybe “amoeboid” is a better description for this microglial phenotype.

Line 422-424: Cite references of studies using these microglial markers (Arginase and inducible nitric oxide synthase) e.g in EAE experiments such as doi.org/10.1186/s12974-016-0730-4

Line 434: make paragraph. Line 435-436: state IL-1β and TNF-α as pro- and IL-10 as anti-inflammatory.

EAE section: Authors are encouraged to provide more information on the severity of the clinical EAE phenotype (such as mean maximal score- MMS; day of disease onset- dDO; area under the curve- AUC etc.) in either main figures or as supplementary material. Currently, there's only a clinical course in Figure 4 showcasing a generic y-axis "increase in paralysis" without a definite scoring scale. Use the following work again as a template doi.org/10.1186/s12974-016-0730-4

Line 468: It is well established that periventricular areas are, amongst many, significant sites of demyelinated plaques. Please add with the appropriate bibliography.

Line 495: Demyelinated plaques and high nuclei area is not the same thing. It is advised to use "inflammatory lesions", characterized by... since authors studied the acute and post-acute phase of EAE.

Line 508 & Figure 4.B: Authors should state if there are any significant differences between groups and the tests used to compare them.

Discussion

Line 543: demyelinating "insult" is not proper description. Employ "after the demyelinating effect of LPC" in the OCSC

Line 545: "lost compactness", instead of lose compaction

Line 602: EAE-challenged not EAE-treated mice again

Line 603: "clear demyelinated plaques in the white matter of the cerebellum at 17 DPI". Authors argue (multiple times in fact throughout the manuscript) that there's an ongoing demyelination within acute inflammatory lesions with plaques already formed. This might be an issue and seems like a bold statement for the animal model used (MOG 35-55), where demyelinating effects are evident in the post-acute, chronic phase, a time point that has not been analyzed (endpoint here is 23 DPI).

Line 626: MS patient application is a long shot perspective. Toxicity and absorption assays should be prioritized and preceded in animals.

Author Response

Reviewer 3

Pinto and colleagues investigate the potential of four green-emitting fluorophores that recognize myelin debris in vitro and in vivo neuroinflammatory milieu. They provide extensive supplementary data on how these dyes were developed accompanied with structural characterization. The one that yielded optimal results (BASHY molecule), was further applied to primary microglia cultures and injected to MOG EAE-challenged mice.

The following comments should be taken into consideration:

We thank again Reviewer 3 for the helpful comments and specific answers are given below:

Abstract

In its current form, it fits more to a review rather than a research article. As clearly stated in the instructions to authors, it should follow the style of structured abstracts, but without headings (i.e background, methods, results and conclusion). Authors are advised to stick to the aforementioned rules and reform it.

As suggested we have rewritten the abstract to follow the aforementioned rules (Line 14).

Introduction

Lines 34-39: Effective remyelination does not depends solely on removal of myelin debris by microglia. Oligodendrocyte apoptosis and endogenous neural precursor cell mobilization and proliferation should also be taken into account.

We agree with the comment and changed to “microglia is one of the important processes that need to occur to promote efficient remyelination” Line 40.

The last 2 paragraphs could be merged into one, describing concisely the purpose of the study.

As suggested, we changed it accordingly.

Line 70: EAE-challenged not EAE-treated mice

We revised the whole manuscript and made the appropriate changes accordingly to the suggestion.

Materials and Methods

Line 162 & 164: Nile Red & Filipin - which source/company?

Filipin was not used in this study, the mistake was corrected Line 187. The company and source of Nile red was added Line 188

Line 168: replace "ImageJ (Fiji Is Just)" with "ImageJ" or "Fiji (Fiji Is Just ImageJ)"

We changed this accordingly Line 200

Results

The results section needs major revisions. First of all, numerical data must be accurately presented (e.g cells/mm2) because the approximate percentages provided (~90%, ~30% , ~6% etc.) currently denigrate the scientific credibility and render this section an extension of a discussion. Additionally, one sentence per result section in the end of the corresponding paragraph, is enough to conclude the findings. All the remainder discursive information, should be integrated into discussion interpreting the findings.

The new quantitative analysis was added to Fig. 2. Quantifications of Iba1+ cells were performed in 3 regions of interest (ROI) within 3 sections/cerebellar White Matter region. The data is presented as the number of cells counted per ROI which was identical for all the quantifications.

Probe candidates 1-4 should be alternatively referred to both text and Figure 1 as e.g test dye 1 (td1), td2 instead of just 1, 2

We revised the whole text and figures and changed it accordingly.

Line 346: This is particularly evident (in or) from... not evidenced by

We agree with the observation and corrected the mistake.

Line 397: "foamy-like microglia" is not commonly used as foamy macrophages. Maybe “amoeboid” is a better description for this microglial phenotype.

The nomenclature foamy was used to distinguish microglia activated (amoeboid) from microglia that is activated, highly phagocytic and show increased levels of intracellular lipids (foamy). To avoid misunderstandings we clarified the definition in the text (Line 455)

Line 422-424: Cite references of studies using these microglial markers (Arginase and inducible nitric oxide synthase) e.g in EAE experiments such as doi.org/10.1186/s12974-016-0730-4

We added the above mentioned referred studies using these microglial markers in EAE experiments. Line 477

Line 434: make paragraph. Line 435-436: state IL-1β and TNF-α as pro- and IL-10 as anti-inflammatory.

As instructed, we made paragraph (line 489) and emphasized the role of IL-1β and TNF-α as pro-inflammatory cytokines and of IL-10 as anti-inflammatory (Line 490)

EAE section: Authors are encouraged to provide more information on the severity of the clinical EAE phenotype (such as mean maximal score- MMS; day of disease onset- dDO; area under the curve- AUC etc.) in either main figures or as supplementary material. Currently, there's only a clinical course in Figure 4 showcasing a generic y-axis "increase in paralysis" without a definite scoring scale. Use the following work again as a template doi.org/10.1186/s12974-016-0730-4

As suggested we introduced in Figure 4 the actual data on clinical score rather than a scheme, we it can be seen mean maximal score- MMS; day of disease onset- dDO; and also the graph with area under the curve- AUC.

Line 468: It is well established that periventricular areas are, amongst many, significant sites of demyelinated plaques. Please add with the appropriate bibliography.

The appropriate bibliography was added, line 529

Line 495: Demyelinated plaques and high nuclei area is not the same thing. It is advised to use "inflammatory lesions", characterized by... since authors studied the acute and post-acute phase of EAE.

We understand the reviewer suggestion. However, in this case, we clearly observed lack of MBP staining in those areas, thus evidencing that occurred demyelination, reason why we used such term.

Line 508 & Figure 4.B: Authors should state if there are any significant differences between groups and the tests used to compare them.

Our results using the EAE model are only demonstrative as we are still optimizing our BASHY. In future experiments, the improved dye will be certainly tested in a higher number of animals using other demyelinated models as the Cuprizone one. Results will be further quantified and compared.

Discussion

Line 543: demyelinating "insult" is not proper description. Employ "after the demyelinating effect of LPC" in the OCSC

The mistake was corrected accordingly Line 600.

Line 545: "lost compactness", instead of lose compaction

The mistake was corrected accordingly Line 601

Line 602: EAE-challenged not EAE-treated mice again

The whole manuscript was revised and the mistake was corrected accordingly

Line 603: "clear demyelinated plaques in the white matter of the cerebellum at 17 DPI". Authors argue (multiple times in fact throughout the manuscript) that there's an ongoing demyelination within acute inflammatory lesions with plaques already formed. This might be an issue and seems like a bold statement for the animal model used (MOG 35-55), where demyelinating effects are evident in the post-acute, chronic phase, a time point that has not been analyzed (endpoint here is 23 DPI).

We agree with the observation. As demyelinating effects are expected only after 20 DPI, we removed demyelinated plaques and rewrite to “early lesioned areas”. Line 673

Line 626: MS patient application is a long shot perspective. Toxicity and absorption assays should be prioritized and preceded in animals.

This is true and will be taking into account in future experiments. To avoid misunderstandings of this final statement, we evidenced our existing hope for this to happen by saying “…either in longitudinal pre-clinical studies using live animals, but hoping to evolve to clinical monitoring of MS patients” Line 693

Round 2

Reviewer 2 Report

The revised manuscript by Pinto et al. supports the conclusions better and has therefore improved. There is one unclear point in the authors' response, though, that requires further explanation. Authors report that they have performed the suggested rotation analysis to test for colocalization by chance (data not provided) and say that they have obtained the exact same results as for the non-rotated colocalization analysis. This result, however, does not support the conclusion that colocalization is real but suggests that colocalization is only by chance. Since it is rather unlikely to obtain the exact same results in this validation analysis, there might be a misunderstanding with the suggested experiment. The proposed validation experiment was to rotate only one of the two image channels by 180 degrees in ImageJ and then use this to colocalize signals in that rotated image (e.g., Bashy) with the signals in the non-rotated original version of the second image channel to be tested for colocalization (e.g., MBP, Qd9, GFAP, and Iba1). For both image channels, authors will need to use the same settings (thresholding, background subtraction, etc.) as for the original colocalization analysis without a rotated channel. If colocalization is real, the results should show a significantly lower amount of colocalization after rotation of one image channel. The results of this validation analysis are important for interpreting the results and should be included in the same figure.

In addition, there were only few minor points where authors should describe some of their findings more explicitly.

  1. Authors should include a description in the results section that they observed many microglia are in control OCSC because of the mechanical injury during slice preparation.
  2. Authors should also provide a more explicit explanation for the finding that more BASHY+/CD80+ cells were found in the cortex compared to the spinal cord in EAE mice.
  3. Authors should also mention more explicitly that the in vivo data in the EAE mice is mostly preliminary and discuss the limitations of these findings in more detail.
  4. In line, 427 authors write 'while showing no affinity for intact myelin'. Writing' low affinity' would better reflect the authors' findings. Similarly, authors should write 'low BASHY colocalization' instead of 'the absence of BASHY colocalization' in line 610.

Author Response

 We thank again the reviewer for these relevant comments and answer accordingly below.

The revised manuscript by Pinto et al. supports the conclusions better and has therefore improved. There is one unclear point in the authors' response, though, that requires further explanation. Authors report that they have performed the suggested rotation analysis to test for colocalization by chance (data not provided) and say that they have obtained the exact same results as for the non-rotated colocalization analysis. This result, however, does not support the conclusion that colocalization is real but suggests that colocalization is only by chance. Since it is rather unlikely to obtain the exact same results in this validation analysis, there might be a misunderstanding with the suggested experiment. The proposed validation experiment was to rotate only one of the two image channels by 180 degrees in ImageJ and then use this to colocalize signals in that rotated image (e.g., Bashy) with the signals in the non-rotated original version of the second image channel to be tested for colocalization (e.g., MBP, Qd9, GFAP, and Iba1). For both image channels, authors will need to use the same settings (thresholding, background subtraction, etc.) as for the original colocalization analysis without a rotated channel. If colocalization is real, the results should show a significantly lower amount of colocalization after rotation of one image channel. The results of this validation analysis are important for interpreting the results and should be included in the same figure.

We thank the reviewer for the question and apologize for the earlier misunderstanding. We performed the suggested rotation analysis following the above guidelines in only one channel. To maintain the same settings, we had to measure again the co-localization of BASHY with Mbp, Qd9, Gfap and Iba1 in specific ROIs, thus a new graph was added to Figure 2, that defines even better the differences between the control and LPC-treated slices. Regarding the new rotation analysis, we provide the data to the reviewer but we do not think that it will improve the manuscript. Indeed, by performing the suggested analysis we found differences, mostly a reduction in the area of co-localization for all markers, except astrocytes, which support that the co-localization is real. We must highlight that the lack of reduction for the co-localization with Gfap is due to a marked staining of this marker in the global picture as explained in the attached document. Changes are highlighted in green (page 5 line 243, Figure 2A).

In addition, there were only few minor points where authors should describe some of their findings more explicitly.

1. Authors should include a description in the results section that they observed many microglia are in control OCSC because of the mechanical injury during slice preparation.

As suggested, this observation was added to the results, new description highlighted in green (page 9, line 405).

2. Authors should also provide a more explicit explanation for the finding that more BASHY+/CD80+ cells were found in the cortex compared to the spinal cord in EAE mice.

In our results for 17 DPI, we found more BASHY+/CD80+ cells in the cerebellum, when compared to the cortex or the spinal cord, while at 23 DPI, following IP injection we have a high amount of BASHY+/CD80+ cells in the cortex that in any other region, potentially due to later cortex alterations. To avoid misunderstandings, we clarified in the text (line 678) that although most EAE-induced lesions occur in both cerebellum and spinal cord, in the spinal cord, the initial evidences of demyelination occur at the lower level before spreading to the upper part, and indeed, was this last region that we used for the cytometry analysis (line 556, 679). This might explain the low number of BASHY+/CD80+ cells in this region, when compared to the number of cells found in the Cerebellum. All changes are highlighted in green.

3. Authors should also mention more explicitly that the in vivo data in the EAE mice is mostly preliminary and discuss the limitations of these findings in more detail.

To avoid misunderstandings, we clarified in the text (line 545, 696) that the in vivo studies were only demonstrative as we were testing, for the first time, this fluorescent dye in vivo. Additional studies are in course to optimize the BASHY probe in order to assess it using different demyelinating models and timepoints. All changes are highlighted in green.

4. In line, 427 authors write 'while showing no affinity for intact myelin'. Writing' low affinity' would better reflect the authors' findings. Similarly, authors should write 'low BASHY colocalization' instead of 'the absence of BASHY colocalization' in line 610.

 In line 431, “showing no affinity” was changed to “showing low affinity” and in line 614 “the absence of BASHY colocalization” was changed to “low BASHY co-localization”. All changes are highlighted in green.
